# Transcranial focused ultrasound stimulation of cortical and thalamic somatosensory areas in human

**Hyun-Chul Kim[1]¤, Wonhye Lee[1], Daniel S. Weisholtz[2], Seung-Schik Yoo[1]***

**1** Department of Radiology, Brigham and Women's Hospital, Harvard Medical School, Boston, Massachusetts, United States of America, **2** Department of Neurology, Brigham and Women's Hospital, Harvard Medical School, Boston, Massachusetts, United States of America

¤ Current address: Department of Artificial Intelligence, Kyungpook National University, Daegu, South Korea
* yoo@bwh.harvard.edu

## Abstract

The effects of transcranial focused ultrasound (FUS) stimulation of the primary somatosensory cortex and its thalamic projection (*i.e.*, ventral posterolateral nucleus) on the generation of electroencephalographic (EEG) responses were evaluated in healthy human volunteers. Stimulation of the unilateral somatosensory circuits corresponding to the non-dominant hand generated EEG evoked potentials across all participants; however, not all perceived stimulation-mediated tactile sensations of the hand. These FUS-evoked EEG potentials (FEP) were observed from both brain hemispheres and shared similarities with somatosensory evoked potentials (SSEP) from median nerve stimulation. Use of a 0.5 ms pulse duration (PD) sonication given at 70% duty cycle, compared to the use of 1 and 2 ms PD, elicited more distinctive FEP peak features from the hemisphere ipsilateral to sonication. Although several participants reported hearing tones associated with FUS stimulation, the observed FEP were not likely to be confounded by the auditory sensation based on a separate measurement of auditory evoked potentials (AEP) to tonal stimulation (mimicking the same repetition frequency as the FUS stimulation). Off-line changes in resting-state functional connectivity (FC) associated with thalamic stimulation revealed that the FUS stimulation enhanced connectivity in a network of sensorimotor and sensory integration areas, which lasted for at least more than an hour. Clinical neurological evaluations, EEG, and neuroanatomical MRI did not reveal any adverse or unintended effects of sonication, attesting its safety. These results suggest that FUS stimulation may induce long-term neuroplasticity in humans, indicating its neurotherapeutic potential for various neurological and neuropsychiatric conditions.

## Introduction

Transcranial focused ultrasound (FUS) techniques allow for non-invasive functional modulation of highly region-specific brain areas by locally delivering low-intensity acoustic pressure waves to the brain [1, 2]. With the ability to reach deep brain areas, which has been challenging

**Data Availability Statement:** The data underlying the results presented in the study are available from the Harvard Dataverse at doi:10.7910/DVN/380AEI [https://dataverse.harvard.edu/dataset.xhtml?persistentId=doi:10.7910/DVN/380AEI].

**Funding:** This work is supported by the Translational Research Institute for Space Health through National Aeronautics and Space Administration (NASA) Cooperative Agreement NNX16AO69A (to SSY). The funders had no role in study design, data collection, data analysis, and decision to publish or preparation of the manuscript.

**Competing interests:** The authors have declared that no competing interests exist.

in other non-invasive brain stimulation (NIBS) techniques, such as transcranial magnetic stimulation (TMS) and transcranial direct current stimulation (tDCS), transcranial FUS has gathered momentum in the neuroscientific community as a new functional neuromodulation modality. With promising safety records gathered from animal models including non-human primate studies [3–5], the appeal of transcranial FUS brain stimulation among healthy humans is trending upward, as demonstrated in studies involving stimulation of cortical somatosensory [6–8], motor [9–11], and primary visual areas [12]. Furthermore, the ability to stimulate deep brain areas has facilitated clinical investigations that examined its effects on suppressing focal epilepsy [13] and on improving disorders of consciousness [14]. Yet, human studies examining stimulation of the thalamus, along with its safety, have been limited.

To achieve neuromodulation, FUS is typically applied in packets of short-duration pulses having a specific pulse duration (PD) at a pulse repetition frequency (PRF), given at an intensity below the level that may elevate tissue temperature [5, 6, 9, 15–17]. Although the exact mechanisms of how ultrasound modulates neural tissue excitability is not yet clearly understood, stimulatory neural tissue responses have been associated with sonication given in short duration (on the order of hundreds of milliseconds) operating through a pulsed mode with a duty cycle greater than 50% (whereby the duty cycle describes the portion of active sonication per stimulation). Use of a pulsing scheme has also shown superior stimulation efficiency over continuous sonication in animal models [18–20]. Through these studies, the choice of sonication parameters has become an important component that may affect stimulation efficiency, demanding in-human evaluation of the effects of PD. Thus, we were motivated to stimulate the primary somatosensory cortex (S1, corresponding to the unilateral hand area representation) and its thalamic projection (*i.e.*, ventral posterolateral nucleus, VPL) in healthy humans, using different PDs (0.5, 1, and 2 ms) at a constant duty cycle of 70%. The stimulatory outcome was assessed from the electroencephalographic (EEG) evoked potentials acquired simultaneously during sonication. Furthermore, to evaluate its safety among healthy individuals, clinical neurological evaluations, EEG, and neuroanatomical MRI were conducted at various time points after the FUS sessions.

In addition to these assessments, one of the important factors to gauge the translational potential of FUS brain stimulation is whether the stimulatory effects can induce durable 'offline' changes in neuronal activities beyond the duration of sonication, ultimately inducing neural plasticity [21–23]. An *in vitro* study on isolated rodent neurons has shown that the enhanced level of evoked potential was sustained more than 4 hours following 40 s-long ultrasound stimulation [24]. In anesthetized rats, FUS stimulation of the somatosensory areas yielded different features in EEG somatosensory evoked potential (SSEP; evoked by the unilateral electrical stimulation of the hind limb), which lasted over 35 min after sonication [25]. A similar observation was reported from ultrasound stimulation of the frontal/supplementary eye fields of non-human primates in modulating the visuomotor activity (anti-saccade behavior performance) [26]. More recently, FUS applied to the medial perforant path of the rat hippocampus generated sustained changes in synaptic connectivity [27]. In humans, theta burst patterned application of transcranial FUS to the motor cortex consistently increased the corticospinal excitability for more than 30 min, as being assessed by conjunctional application of TMS [28], and an altered state of the default mode network was reported after application of FUS to the right prefrontal cortex [29]. These studies suggest the potential for inducing durable functional changes of the brain that outlast the sonication.

While EEG and behavioral performance relating to brain stimulation provide information on neuronal responses with limited neuroanatomical information, acquisition and analysis of resting-state functional magnetic resonance imaging (rs-fMRI) offer an excellent insight on brain-wide functional connectivity (FC) by measuring temporal correlations of spontaneous

blood oxygenation level-dependent (BOLD) signals across brain regions [30, 31]. The technique is completely non-invasive and has been widely used to detect resting-state FC (rsFC) changes associated with pathological conditions or various neurological interventions [32–36]. In macaques, rsFC analysis was adopted to demonstrate the durable off-line neuromodulatory changes induced by FUS stimulation transcranially delivered to the supplementary motor area (SMA) [3]. One of important aims of this study was to evaluate the rsFC changes induced by FUS stimulation of the thalamic VPL.

## Materials and methods

### Study overview and study participants

The study consisted of two segments. The first segment evaluated FUS-evoked EEG potentials (referred to as 'FEP' herein) and off-line rsFC changes associated with the stimulation, along with comprehensive safety assessments. As a few participants perceived auditory tones during the FUS stimulation (described in 'Subjective reporting' of the Results section), the second segment examined EEG auditory evoked potentials (AEP) in response to audible tones from a separate group of participants. The first segment was approved by the Institutional Review Board (IRB) of Brigham and Women's Hospital (Protocol#: 2019P001666), accompanied by the Food and Drug Administration (FDA) determination of the device safety as Non-Significant Risk category. A separate IRB approval was obtained to measure AEP (Protocol#: 2121P002025). All participants gave written informed consent before the initiation of each study.

Primary inclusion criteria were healthy male and female adults aged 21–45. In the first study segment, a total of 32 individuals responded to a public posting recruitment, 13 of whom gave a written consent to participate. Five of the consented individuals withdrew prior to participating in FUS sessions and the remaining eight completed the study (three females, age = 32.1 ± 6.0 y.o., range: 23–42, mean ± standard deviation). For AEP measurement, eight (four females, age = 32.3 ± 8.3 y.o., range: 21–44) completed the study out of 16 individuals who responded to the recruitment posting. The age of the participants between both protocols was not different (*t*-test; two-tailed, *p* = 0.97), and all were right-handed based on the Edinburgh handedness inventory [37]. Participants with a history of neurological (including peripheral nerve diseases) or psychiatric conditions, active use of sedatives, analgesics, and pharmacological agents that may affect brain function, were not included in the study. Participants who had contraindications for computed tomography (CT) and magnetic resonance imaging (MRI) scans were excluded from the first study segment.

The experimental procedure is described in Fig 1. After providing written consent in the presence of a physician study staff (Visit #0), the participants received MRI and CT scans of their head as well as the first clinical neurological evaluation (NE; Visit #1). The NE, conducted by a licensed neurologist, assessed the participants' mental status, deep-tendon reflexes, gait/stance, cranial nerve II-XII function, motor, somatosensory, and cerebellar functions. The MRI included both anatomical and functional MRI to identify individual-specific locations of the somatosensory circuits of their non-dominant hand (*i.e.*, left hand), along with rs-fMRI to obtain baseline rsFC information. Then, acquired MRI data were processed for image-guided FUS stimulation.

Following Visit #1, two separate FUS sessions (Visits #2 and #3; conducted about a week apart, 7.9 ± 2.5 days) were administered to stimulate the S1 and VPL, respectively in sequence. A time gap of one week between two FUS stimulation sessions was introduced to allow for the observation of any unexpected safety-related side effects from participants while allowing them enough time to consider retraction from the study. Stimulation of both S1 and VPL

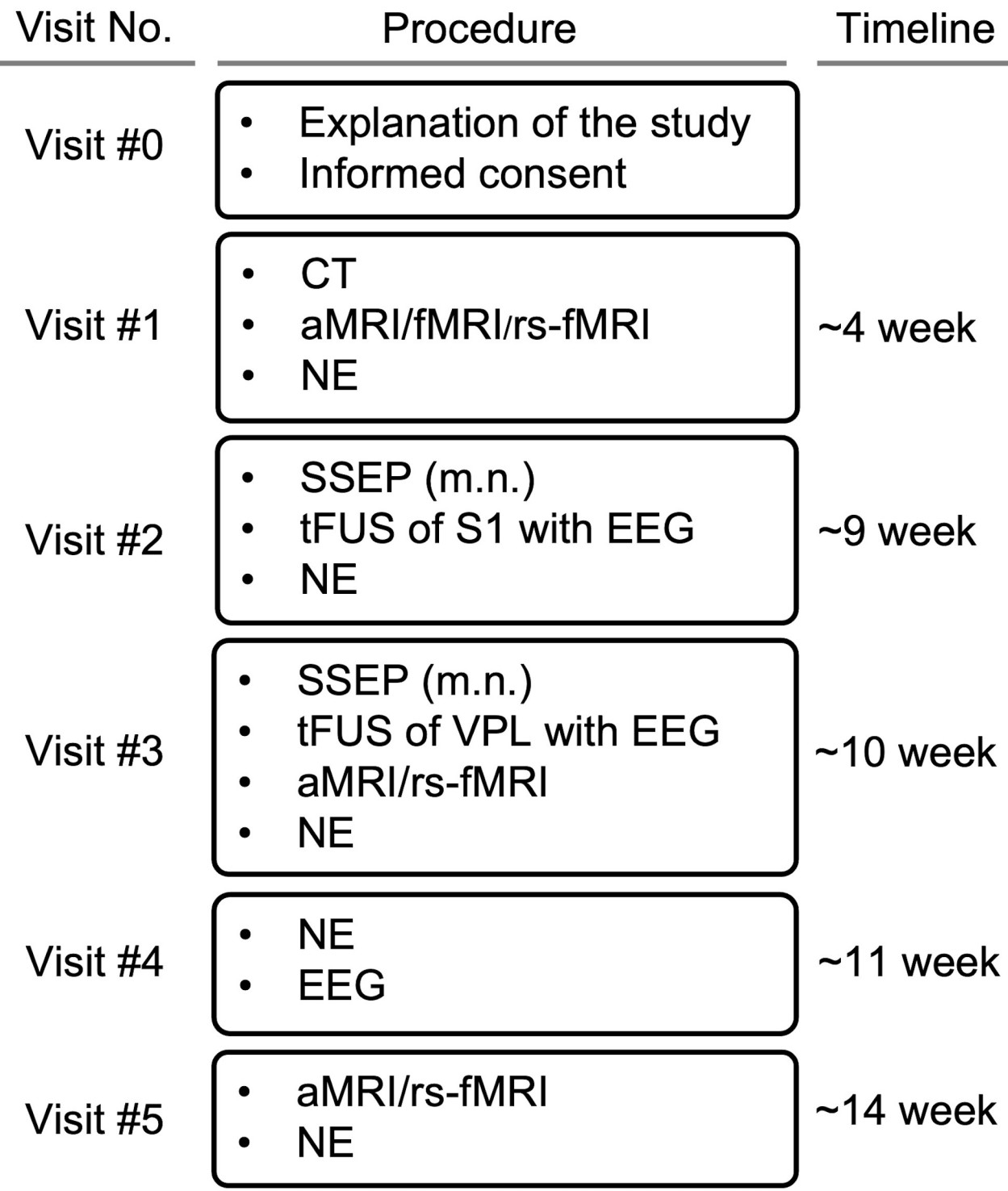

**Fig 1. Experimental procedure.** Acronyms used—CT: Computed tomography, aMRI: Anatomical magnetic resonance imaging, fMRI: Functional magnetic resonance imaging, rs-fMRI: Resting-state functional magnetic resonance imaging, NE: Neurological evaluation, SSEP: Somatosensory evoked potentials evoked by median nerve (m.n.) stimulation, FEP: FUS stimulation-mediated evoked potentials, EEG: Electroencephalography. S1: Primary somatosensory cortex, VPL: Ventral posterolateral nucleus.

within a single session was not conducted to avoid fatigue from long experimental time, which would take far more than an hour (~ 2 hrs) involving the needs for separate image-guidance procedures. Prior to each FUS session, SSEP elicited by electrical stimulation of the left wrist was measured. The FEP responses to FUS stimulation were also recorded using the same EEG montage. Upon completion of each FUS session, a NE immediately followed, with the inclusion of a brief inquiry of subjective sequelae of the stimulation. The participants were not informed regarding the presence, including types, of any sensation prior to the sonication while any verbal descriptors that may bias the perception were not used by the study staff throughout the study.

On Visit #3, anatomical and rs-fMRI were administered to examine any anatomical abnormalities or off-line rsFC changes that may transpire shortly after the VPL stimulation (82.5 ± 40.3 min gap, ranges 40–140 min). One week (7.4 ± 3.1 days) after Visit #3, participants underwent EEG assessment (simplified—$F_3/F_4$; $C_3/C_4$; $T_3/T_4$; $P_3/P_4$ montages) and another NE (Visit #4). About four weeks (26.1 ± 4.9 days) after Visit #3 (VPL stimulation), the study segment was concluded with a final follow-up NE and MRI session (Visit #5) which included the acquisition of the third off-line rs-fMRI data.

All anatomical MRI and CT images from the participants were evaluated by licensed radiologists who were blinded to the study. The participants who were recruited for the AEP measurement underwent a single EEG session without undergoing neuroimaging and NE, at least a day after providing written consent.

## MRI/CT scans and sonication target

Prior to the MRI/CT session, four doughnut-shaped multimodality fiducial markers (Beekley Medical, Bristol, CT) were attached to the skin in nonplanar locations on the forehead and behind each ear. Subject-specific skin blemishes, wrinkle lines, and cutaneous veins were used as reproducible spatial coordinates for the spatial registration between the image data ('virtual space') and the head ('physical space') [7].

High-resolution anatomical MRI data (3 Tesla, Skyra, Siemens, Munich, Germany) were obtained using 3D magnetization-prepared rapid gradient-echo (MP-RAGE) sequence (repetition time (TR)/echo time (TE) = 2,300/2.3 ms; field-of-view (FoV) = 250 mm × 250 mm; voxel size = 0.98 mm × 0.98 mm × 1.0 mm with no gaps between the slices; flip angle (FA) = 8˚; 176 sagittal slices), covering the entire head. Gradient echo (single-shot) echo-planar imaging (EPI) sequence (TR/TE = 2,000/40 ms, FA = 90˚, FoV = 240 × 240 mm$^2$, voxel size = 3.75 mm × 3.75 mm × 4 mm with 1-mm slice gaps, 22 axial slices, whole brain coverage excluding the cerebellum) was used to obtain the BOLD fMRI data. To identify the somatosensory areas that innervate the unilateral hand, the subject clenched their non-dominant (left) hand at ~1 Hz during three 20 s task blocks interleaved with four resting blocks of equal duration. Using the same sequence and volume coverage, rs-fMRI data were also acquired for 5 min (*i.e.*, 150 fMRI volumes with TR = 2 s). Subjects were asked to relax and lie still in the scanner while remaining calm and awake with their eyes closed. High-resolution $T_1$-weighted (TR/TE = 400/9.4 ms, FoV = 240 mm × 240 mm, voxel size = 0.47 mm × 0.47 mm × 4.0 mm, FA = 65˚, 22 axial slices) and $T_2$-weighted (TR/TE = 4,340/100 ms, FoV = 195 mm × 240 mm, voxel size = 0.47 mm × 0.47 mm × 4.0 mm, FA = 150˚; 22 axial slices) images were also acquired from the same brain volume as the EPI acquisition. An fMRI activation map was created by deriving a voxel-wise degree of synchrony of BOLD time-series with respect to a task-specific canonical hemodynamic response after motion correction and spatial smoothing (3D Gaussian kernel with a full-width at half-maximum [FWHM] size of 6 mm) using SPM8 software (Wellcome Department of Imaging Neuroscience, University College London, London, UK).

A clinical CT scanner (Aquilion ONE, Toshiba, Japan) was used to acquire cranial information needed for estimating the *in situ* acoustic intensity at the focus *via* numerical simulation (FoV = 240 mm × 240 mm, voxel size = 0.47 mm × 0.47 mm × 0.5 mm, entire head coverage). When acquiring CT data, a 3 mL vial containing water was placed in the FoV and was used as a reference to calibrate the image intensity in Hounsfield Units (HU; water is 0). Individual CT data were screened for the presence of large intracranial calcification (size greater than wavelength of the FUS, *i.e.*, 5 mm, with HU > 1000 [38]), which may absorb/distort incident acoustic energy. CT, to limit unnecessary radiation exposure, was acquired only once from each participant.

Subsequently, CT and MRI data (both anatomical and functional) were spatially co-registered (Amira, Thermo Fisher Scientific, Waltham, MA). Then, the spatial coordinates for FUS stimulation, all from the right hemisphere, were individually identified from fMRI maps. The S1 target was determined from the local activation maxima observed in the postcentral gyrus while the VPL was identified from the local maximum coordinate anterior to the ventral lateral nucleus, all cross-referenced to the anatomical MR images.

## FUS device for image-guided stimulation

We used an investigational in-house device to deliver FUS to the target somatosensory areas under image-guidance (Fig 2A). Sinusoidal electrical signals used to drive the FUS transducer were generated by two sets of function generators (33500B, Keysight, Santa Rosa, CA). The signal was amplified by a 10 W capacity power amplifier (Sonomo 500, Electronics and Innovation, Rochester, NY) with impedance matching (JT series, Electronics and Innovations). A custom-made independent voltage follower and a circuit breaker (CleCell, Seoul, Korea) were added to monitor the output from the function generator and to cut power to the linear amplifier in case the input signal exceeded the user-defined voltage level (and associated acoustic intensity output).

Two different types of gas-matrix piezoelectric FUS transducers (GPS200-D40-FL57, referred to as 'D40' and GPS200-D90-FL130, referred to as 'D90', Ultran Group, State College, PA) were mounted on a headgear (Fig 2B). The focus was formed 31 mm from the exit plane of the D40 transducer (used for S1 stimulation). The D90 transducer, having a focal length of 84 mm, was used to sonicate the VPL. Calibration of acoustic power (from the input sinusoidal wave voltages) and measurement of the pressure profile at the focus were performed in degassed water using calibrated hydrophones (HNR-500 and HNC-0200, Onda Corp, Sunnyvale, CA) mounted on a robotic, 3-axis linear stage (Bi-Slides, Velmex Inc., Bloomfield, NY). The size of the focus, an elongated ellipsoidal shape defined at the area bound by FWHM intensity, was 10 mm in diameter and 56 mm in length (D40, Fig 2C) and 12 mm in diameter and 73 mm in length (D90, Fig 2D, all indicated with black dotted lines). At 90%-maximum intensity, which has shown to closely estimate the areas of neuromodulatory effects in rodents [39], the focal size was 3 mm in diameter and 13 mm in length (D40, Fig 2C) and 4 mm in diameter and 21 mm in length (D90, Fig 2D, indicated with white dotted lines).

A compressible cone-shaped polyvinyl alcohol (PVA) hydrogel (two freeze-thaw cycles, 9% weight per volume in degassed water, 341584, MilliporeSigma, St. Louis, MO) was made in-house [39], with its thickness adjusted to fill the gap between the transducer and the scalp for adequate acoustic coupling. Ultrasound gel (Aquasonic, Parker Laboratories, Fairfield, NJ) was also applied between all interfaces (*i.e.*, the PVA coupler and scalp/transducer surfaces).

The FUS transducer was placed over the subject's head using an image-guided technique described in detail elsewhere [7, 8, 40]. The process included the determination of the desired entry/target point, sonication angular orientation, and real-time display of spatial error

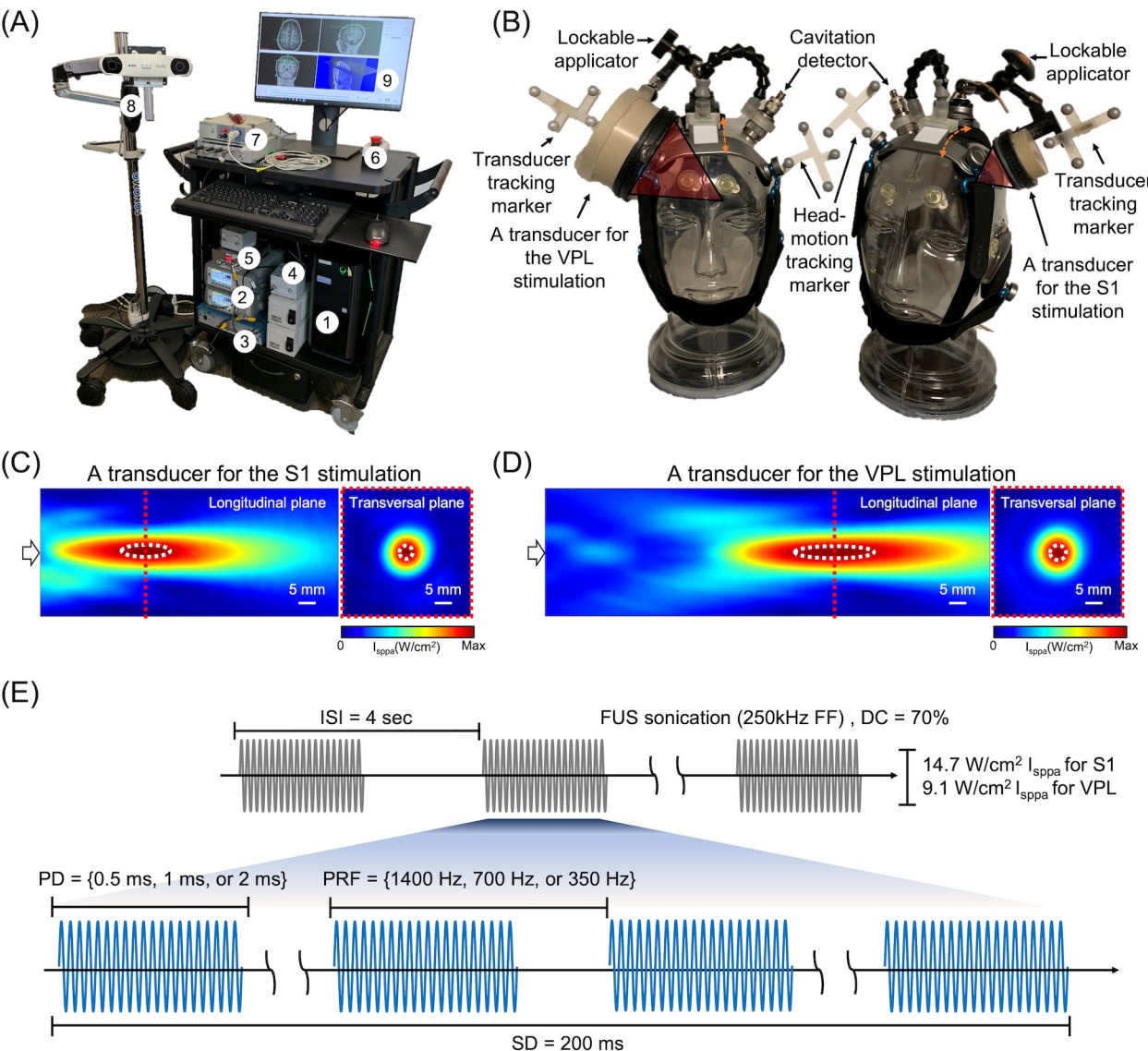

**Fig 2. Schematics of experimental settings.** (A) FUS system that consists of (1) a control computer, (2) function generators, (3) amplifier, (4) impedance matching box, (5) voltage following circuit, (6) emergency stop button, (7) EEG data acquisition and median nerve stimulation system, (8) image-guidance infrared camera, (9) display monitor. (B) FUS transducer headgear for the S1 and VPL stimulation. (C) the spatial profiles of acoustic intensity of transducers for S1 stimulation and (D) VPL stimulation. The white arrows represent the sonication direction. Dotted white lines depict the regions of 90%-maximum of the intensity. (E) illustration of acoustic parameters: inter-stimulation interval (ISI), fundamental frequency (FF), duty cycle (DC), pulse duration (PD), pulse repetition frequency (PRF), sonication duration (SD), spatial-peak pulse-average intensity ($I_{sppa}$).

between the tracked focal location and the stimulation target. The tracking camera (Northern Digital, Inc., Waterloo, Ontario, Canada; Fig 2A) was mounted to an articulated arm for maneuvering. The virtual space of the head was co-registered to its corresponding real space, represented by four donut-shaped markers placed on the skin during the acquisition of initial CT and anatomical MRI. Once registered, the head motion tracker (Fig 2B) was used to track the orientation and location of the transducer (and its acoustic focus) with respect to the head motion. The spatial accuracy of the image-guided navigation system was 0.57 ± 0.14 mm (measured from plastic head phantom), which is far smaller than the size of the acoustic focus.

Once the operator aligned the transducer to place the acoustic focus to the desired target, the transducer was locked in place and moved with the head in synchrony. Care was given to place the transducer so that its sonication path was as perpendicular as possible to the skull tangential surface to minimize the effects from refraction due to the skull [41]. Prior to positioning the transducer/coupler over the scalp, hair was carefully parted away from the center of the sonication path and degassed ultrasound gel was applied over the scalp.

For the FUS stimulation, a fundamental sinusoidal frequency of 250 kHz was used. Three types of stimulation trials using different PDs (0.5, 1 and 2 ms), each 200 ms long, were delivered based on our experience in stimulating small/large animals and humans [7, 8, 18, 19, 40]. To maintain a 70% duty cycle across the varying PDs, the PRF was adjusted to 1,400, 700, and 350 Hz respectively for each stimulation trial (The graphical illustration of the pulse scheme is shown in Fig 2E). Stimulation trials were administered 4 s apart whereby the computer generated 80 trials per PD condition in a randomized and balanced fashion with the inclusion of a passive sham condition (*i.e.*, no sonication), which rendered a FUS session lasting ~21 min (1,280 s). The free-water (*i.e.*, un-derated) spatial-peak pulse-average intensity ($I_{sppa}$) was 14.7 W/cm$^2$ for S1 and 9.1 W/cm$^2$ $I_{sppa}$ for VPL stimulation.

## Semi real-time numerical simulation of acoustic propagation

We implemented an acoustic simulation algorithm in the control computer that executed the image-guidance and sonication. The software was installed in a high-performance computer (Intel Core i7-9700 CPU @ 3.60 GHz, 64 GB memory, Microsoft Windows 10, 64 bits) having a graphics processing unit (GPU, GeForce RTX 2080 Ti, NVIDIA, Santa Clara, CA). The finite-difference time-domain formulation was adopted to model the transcranial propagation of acoustic waves from a single-element FUS transducer to estimate the location and intensity of the acoustic focus at the target [41]. Material properties of water (speed of sound = 1,500 m/s, density = 1,000 kg/m$^3$, attenuation coefficient = 0 Np/m) and the human skull (attenuation coefficient = 33 Np/m at 270 kHz) [42, 43] were used as inputs in the simulation. Simulation was performed with a discretized time interval of 0.1 μs, using a spatial resolution of 0.5 mm covering the volume encompassing both the head and transducer (25 × 25 × 15 cm, Left-right × Anterior-Posterior × Superior-Inferior directions). As the speed of sound in the bone and bone density play an important role in the simulation, the speed of sound was modeled in two steps (2,140 m/s for the bone having HU 1–1,000, and 2,384 m/s in bone with >1,000 HU) [43]. The bone density was modeled at linear increment from 1,500 to 2,190 kg/m$^3$ (in a HU range of 1–1,000) and 2,190 kg/m$^3$ for tissue having HU > 1,000 [42]. The accuracy of the simulation algorithm was evaluated using *ex vivo* adult human skull samples (*n* = 3; see S1 Appendix. "**Validation of numerical simulation**" section for details, and S1 Fig), and the averaged error in estimating the acoustic focal location/dimension (< 5 mm) and *in situ* pressure (< 10%) suggested its reliable operation. The use of GPU-based parallel computation enabled semi real-time (less than 40 s) feedback to the operator. If the location of the focus estimated through simulation deviated greater than 4 mm from the intended target, the transducer was repositioned. After locking the transducer location in place with respect to the headgear, the FUS session commenced.

## Retrospective numerical evaluation of acoustic propagation

Despite the use of image-guidance and simulation-assisted initial placement of the transducer, imperfection of the manual placement of the transducer and the acoustic propagation through the skull may introduce unintended spatial error and additional intensity attenuation. Thus, we acquired the sonication geometry upon completion of each stimulation session and re-

evaluated the acoustic propagation profile within individuals' calvarium through the simulation using the same simulation platform deployed for image-guidance. The spatial error between the geometric target coordinates and the estimated maxima of the acoustic focus in the brain was calculated.

### Passive cavitation detection

Despite the extremely low likelihood of cavitation, we also monitored the presence of cavitation using Acoustic Emission Detection (AED) during the transcranial application of FUS. To detect any cavitation events, a broadband ultrasound transducer (0.5 MHz, V318-SU, Olympus NDT, Waltham, MA) was used as an AED sensor, placed on the surface of the forehead toward the site of target focus, with application of ultrasound gel for acoustic coupling (Fig 2B). The emission signals were displayed in the frequency domain using a real-time spectrum analyzer with built-in 20 dB preamplifier (SSA3021X-TG, Siglent, Transcat, Rochester, NY) and cavitation events that may be associated with potential tissue damage (broadband frequency emission other than sonication frequency [44, 45]) were monitored. Although a multi-array detector configuration may provide information on the potential location of cavitation [46], we adopted a single detector element for its small size that can be placed between the EEG electrodes. The device performance was evaluated by detecting the intentional cavitation events by sonicating microbubbles (Definity, Lantheus, Billerica, MA; prepared in 0.1 mL volume concentration in normal saline) (S2 Fig).

### Measurement of reference SSEP and FUS-mediated evoked potential (FEP)

To be used as a reference EEG feature, SSEP from electrical stimulation of the median nerve of non-dominant (left) wrist areas were measured (constant electrical current of $15.9 \pm 1.5$ mA, range = 13–19 mA, 50 μs duration, 100 stimulations with an interstimulus interval of 1 s). EEG recordings were acquired from four adhesive electrodes (Covidien, Mansfield, MA) placed on the $F_3$, $F_4$, $P_3$ and $P_4$ montage (BioAmp, ML408; ADInstruments, Colorado Springs, CO) using an acquisition software (PowerLab; ADInstruments), with a reference electrode (ref) placed on the right mastoid. The gap between the electrodes allowed for the access of both the FUS transducer and PVA gel. Voltage measurement between each pair was defined by ($F_3$-ref)-($P_3$-ref) and ($F_4$-ref)-($P_4$-ref), each representing EEG signals from $F_3$-$P_3$ and $F_4$-$P_4$.

### Assessment of presence of EEG signal artifact due to sonication

To assess the presence of sonication-related artifacts in the recorded FEP, we measured EEG signals from a hydrogel phantom simultaneously with application of FUS. A hydrogel phantom was prepared by dissolving PVA in 150 mL of distilled water (7% weight/volume concentration), which was molded in a 14 cm diameter petri dish and subjected to 2 freeze-thaw cycles to make it in the form of a 10 mm-thick disk. After placing the gel phantom over a 7 mm-thick rubber pad for acoustic absorption, the transducer was positioned over the surface of the phantom along with acoustic coupling gel. Then, EEG electrodes were placed around the transducer mimicking the location used *in vivo* experiment. The same experimental procedures as the *in vivo* experiment were used to separately acquire the EEG data from two transducers.

### AEP measurement

To probe the effects of auditory phenomena and their potential contribution to the observed FEP, AEP was separately measured from a group of participants using the same experimental

procedure as the SSEP/FEP measurement (*i.e.*, same number of stimulation trials, including the trials having a 'no sound' condition, and presentation interval/randomization/balancing). Instead of applying FUS, 200 ms-long auditory tones (sinusoidal) which mimicked the auditory phenomena (associated with the pulse repetitions [47]) were given mono-/binaurally at 1,400, 700, and 350 Hz using earbud headphones (JBL, Los Angeles, CA). Monaural auditory stimulation was also provided to simulate the condition whereby a few individuals reported hearing tones unilaterally during FUS sessions (described in the **Results** section). Tonal intensity for AEP (59.9 ± 7.2 dBA, measured 10 times across the different frequencies using a decibel meter (R8060, REED Instruments, Wilmington, NC)) was much greater than the level reported by the subjects participated in the FUS sessions. For the monaural condition, the opposite ear to the sound stimulation was blocked with an ear plug (Classic, 3M, London, Canada) to reduce potential confounds from ambient noise.

## EEG data analysis

The mains filter was first applied to EEG data to remove alternating current (AC) power line noises and band-pass filtered (0.3–100 Hz). Then, the data were segmented in a time-locked fashion covering 200 ms before and 800 ms after sonication onset. Occasional signal saturation (|amplitude| > 100 μV) and posture-related large deflections in EEG signals were subsequently excluded. The signal fluctuations associated with eye blinking were corrected by removing 0–8 Hz electrooculographic frequency band using a regression approach [48]. In addition, low-frequency signal fluctuations associated with respiratory/cardiac signals, which appeared simultaneously in both hemispheres, were removed using nonlinear drifting correction through Discrete Cosine Transform (DCT) filter (with five basis functions). The resulting signal was baseline (a direct current) corrected with respect to -200–0 ms. The selected trials and corresponding EP were averaged and subjected to group-level analysis using paired and two-sample *t*-tests with random permutations (*n* = 10,000) to compare changes in EEG amplitude between hemispheres and PD conditions, respectively. Kruskal-Wallis one-way analysis of variance (ANOVA) were conducted followed by *post-hoc* analysis for multiple comparisons to compare changes in EEG amplitudes across the four trial conditions (significance defined $p < 0.005$). Data processing and statistical analyses were performed using MATLAB (MATLAB 2020b, Mathworks, Natick, MA).

## rsFC analysis

The rs-fMRI data were processed using SPM8 software (https://www.fil.ion.ucl.ac.uk/) with the exclusion of the first 6 s dummy volumes to allow $T_1$-signal equilibration. Slice timing was corrected, and motion-related images were realigned. The rs-fMRI data were normalized to the Montreal Neurological Institute (MNI) space (3 mm × 3 mm × 3 mm), and subsequently underwent spatial smoothing (3D Gaussian kernel with a FWHM size of 8 mm). Physiological noise correction was performed using the aCompCor method with five principal components (PCs) obtained from the cerebrospinal fluid (CSF) and white matter (WM) regions, respectively. More specifically, *a priori* maps with a probability greater than 0.7 for the CSF and 0.9 for the WM areas in the MNI space, available in MRIcron software [49], were used to estimate the PCs of the CSF and WM, respectively. Then, nuisance noises were removed from BOLD signals *via* regression analysis using six motion parameters and their first derivatives. The resulting BOLD signals were bandpass filtered 0.01–0.1 Hz, and frame-wise displacement (FD) estimation on temporal head movement was conducted using six motion parameters obtained from the 'Realignment' step to exclude spurious motion-related BOLD signal (FD score > 0.5) [50].

The rsFC map was obtained by calculating temporal correlations between BOLD signals from the sonicated area and other brain regions. To derive brain-wide FC with respect to the sonicated areas, 'seed' regions-of-interest (ROIs) each having a sphere of 6 mm in radius (33 voxels) were defined at the sonicated S1 and VPL areas. Then, the average BOLD time series across voxels within each ROI were used to calculate Pearson's correlation coefficients (CCs) with the BOLD signals from all possible pairs of in-brain voxels (other than the seed regions). The CCs values representing the levels of rsFC between pairs of voxels were then normalized *via* Fisher's r-to-z transformation and were subjected to a one-way repeated measures ANOVA followed by *post-hoc* analysis for multiple comparisons among three visits (*i.e.*, Visits #1, #3, and #5; significant level $p < 0.05$ with Bonferroni correction).

## Results

### Subjective reporting

Subjective descriptions of the sensations perceived during FUS stimulation are listed in Table 1. A fraction of the subjects (three from the S1 stimulation and one from the thalamic stimulation; $n = 8$) reported intermittent, but clear, perception of tactile sensations in either the left or both hands. The descriptors included tingling/vibratory sensation, and involuntary muscle movement. One individual who received thalamic stimulation reported tactile sensation contralateral to sonication in addition to ipsilateral vibratory sensation at the scalp. A greater portion of the subjects reported hearing auditory tones ($n = 6$ during S1 stimulation and $n = 5$ during thalamic stimulation), being described as faint beeps (barely audible) having different tones heard through one or both ears. Not all subjects reported hearing the sound and none of these sensations were painful/uncomfortable.

**Table 1. Subjective reporting on tactile/auditory sensations and summary of individual numerical estimation of acoustic intensity and spatial errors of FUS stimulation based on numerical simulation.** Spatial errors were estimated using the focal coordinates obtained from the simulation and those obtained from the intended stimulation target.

| ID | S1 stimulation | | | | | VPL stimulation | | | | |
|---|---|---|---|---|---|---|---|---|---|---|
| | Tactile sensation (Yes/No) | Auditory sensation (Yes/No, Side) | *in situ* $I_{sppa}$ (W/cm$^2$) | MI | Spatial error (mm) | Tactile sensation (Yes/No) | Auditory sensation (Yes/No, Side) | *in situ* $I_{sppa}$ (W/cm$^2$) | MI | Spatial error (mm) |
| 1 | No | Yes, Right | 2.4 | 0.5 | 4.6 | No | Yes, Both | 2.7 | 0.6 | 2.9 |
| 2 | Yes[a] | No | 5.0 | 0.8 | 1.9 | No | No | 4.3 | 0.7 | 2.0 |
| 3 | No | Yes, Both | 3.7 | 0.7 | 3.0 | No | Yes, Both | 5.3 | 0.8 | 1.4 |
| 4 | No | Yes, Both | 2.7 | 0.6 | 1.9 | No | No | 4.2 | 0.7 | 3.2 |
| 5 | Yes[b] | Yes, Left | 3.0 | 0.6 | 3.0 | Yes[d] | Yes, Left | 3.3 | 0.6 | 1.7 |
| 6 | No | Yes, Left | 5.9 | 0.8 | 0.9 | No | Yes, Left | 2.9 | 0.6 | 2.1 |
| 7 | No | Yes, Left | 8.6 | 1.0 | 1.0 | No | Yes, Both | 2.6 | 0.6 | 0.9 |
| 8 | Yes[c] | No | 1.7 | 0.5 | 2.9 | No | No | 1.9 | 0.5 | 1.4 |
| | | Mean | 4.1 | 0.7 | 2.4 | | Mean | 3.4 | 0.6 | 2.0 |
| | | s.d. | 2.3 | 0.2 | 1.2 | | s.d. | 1.1 | 0.1 | 0.8 |

$I_{sppa}$: Spatial-peak pulse-average intensity, MI: Mechanical index, S1: Primary somatosensory area; s.d.: Standard deviation, VPL: Ventral posterolateral nucleus.

[a] Tingling sensation from the left wrist and hand (dorsal thumb, medial wrist area).

[b] Tingling/movement sensations from the right and left palm side with dominance in the right palm.

[c] Vibrating sensation on the palmer side from both hands. Report of a sensation 'within the head.'

[d] Tingling and muscle movement sensation from the left palm side and wrist, and sometimes from both hands. Tactile sensation at the scalp ipsilateral to sonication.

## Numerical estimation of acoustic propagation

The results of the simulated acoustic propagation profile within individuals' calvarium (S3 Fig; shown at the coronal plane intersecting the simulated maximum pressure) yielded an average *in situ* $I_{sppa}$ of 4.1 ± 2.3 W/cm² for the S1 stimulation and 3.4 ± 1.1 W/cm² for the VPL stimulation at the intended target (Table 1). Considering the 70% duty cycle, *in situ* spatial-peak temporal-average intensity ($I_{spta}$) were 2.9 ± 1.6 W/cm² for the S1 stimulation and 2.4 ± 0.8 W/cm² for thalamic stimulation whereby these intensities would not significantly raise the tissue temperature (< 0.5˚C) based on previous numerical estimations that utilized similar pulsing schemes [7, 8, 19]. The estimated *in situ* mechanical indices (MIs) were 0.7 ± 0.2 and 0.6 ± 0.1 for the S1 and VPL stimulation respectively, all under the threshold for mechanical effect (FDA regulatory limit of 1.9 for ultrasound imaging of most parts of the adult body). The averaged spatial error between the intended target coordinates and the estimated maxima of the acoustic focus was 2.4 ± 1.2 mm and 2.0 ± 0.8 mm for the stimulation of S1 and VPL, respectively. The presence of tactile/auditory sensations was not correlated with *in situ* $I_{sppa}$ and the spatial error (Multiple regression $R^2$ = 0.10 and 0.13, *p* = 0.77 and *p* = 0.70, respectively). We note that exceptionally high intensity (8.6 W/cm²) from one subject (ID7; Table 1) contributed to greater group-averaged *in situ* $I_{sppa}$ in case of S1 stimulation than the thalamic stimulation. When the values from ID7 were excluded, the *in situ* $I_{sppa}$ was not different between stimulation of S1 and VPL (3.5 ± 1.5 and 3.5 ± 1.2 W/cm², respectively, paired *t*-test, *p* = 0.96).

## SSEP and FEP comparisons across PD conditions

The time-locked SSEP (covering 200 ms before and 800 ms after the stimulation onset) from electrical stimulation of the left wrist showed a distinct N30 component from the right hemisphere ($F_4$-$P_4$, in blue Fig 3A) as well as a slightly smaller N30 component from the ipsilateral

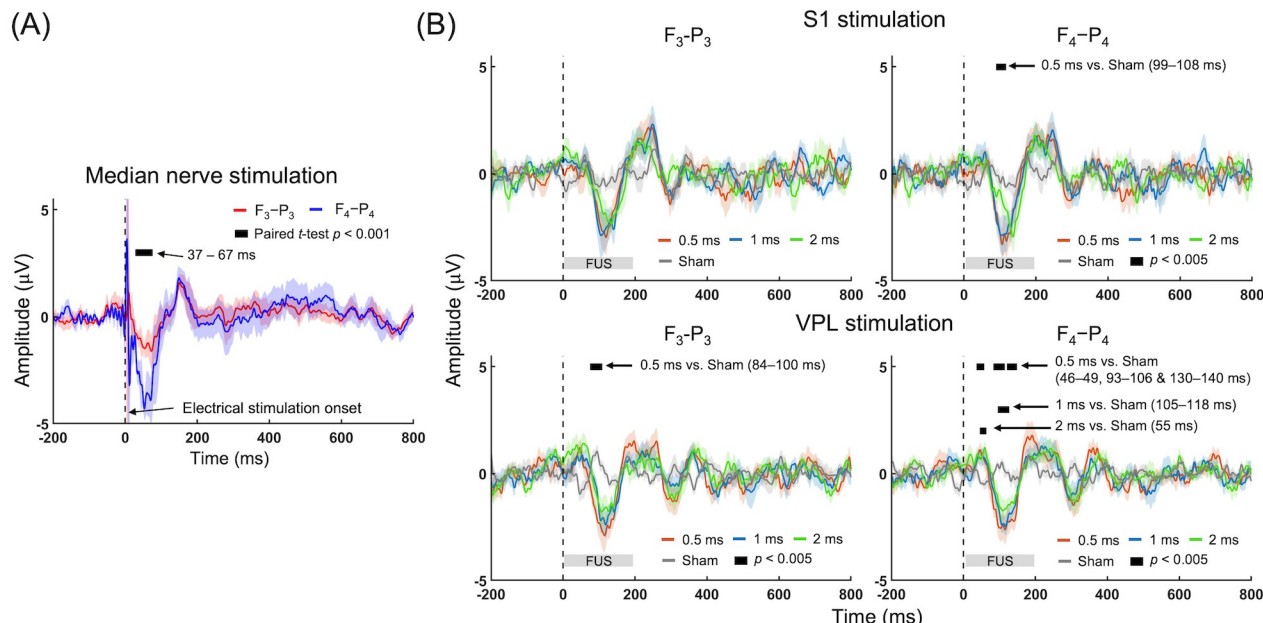

**Fig 3. Group-averaged SSEP and FEP.** (A) SSEP from the left (red: $F_3$-$P_3$) and right (blue: $F_4$-$P_4$) hemispheres. (B) FEP acquired from different PD conditions. Kruskal-Wallis one-way ANOVA (*p* < 0.005) was conducted to compare FEP obtained from the four experimental conditions followed by *post-hoc* analysis for multiple comparisons (*p* < 0.005). The solid lines and shaded areas indicate group-average and standard errors, respectively.

side (F$_3$-P$_3$, in red Fig 3A). This indicates that unilateral nerve stimulation activates the sensory areas of both brain hemispheres, with greater involvement contralateral to the stimulation.

The group-averaged number of FEP data sets analyzed after excluding signal saturation and motion-related artifacts were 68.0 ± 8.2, 68.0 ± 8.3, 68.9 ± 6.4 (all out of 80) for the 0.5, 1, and 2 ms PD conditions respectively in the S1 stimulation whereas the equivalent values were 68.9 ± 11.0, 68.4 ± 10.7, and 70.6 ± 7.3 in the case of VPL stimulation. There was no data selection bias across the experimental conditions (Kruskal-Wallis one-way ANOVA, $F(2,23) =$ 0.03, $p = 0.97$, S1 stimulation; $F(2,23) = 0.12$, $p = 0.89$, VPL stimulation).

Although tactile sensations were not felt across all participants, FEP features were observed across all subjects. Group-averaged amplitudes of FEPs to the stimulation of the S1 and the VPL are shown in Fig 3B across the PD conditions. FUS elicited distinct signal features that shared similarities to the SSEP evoked by median nerve stimulation (*e.g.*, negative peaks of 37–67 ms latency; Fig 3A). The FEP amplitude measured from the right hemisphere (ipsilateral to stimulation of the S1) revealed that the 0.5 ms PD condition generated greater absolute FEP amplitudes than those obtained from the sham condition in the time segments of 99–108 ms (Kruskal-Wallis one-way ANOVA, $F(3,31) = 12.8$–14.7, $p < 0.005$, followed by Tukey-Kramer *post-hoc* analysis $p < 0.005$). Conversely, the FEP measured from the contralateral to FUS stimulation of the S1 were not different across the PDs. The sham condition did not yield any FEP peaks.

The VPL stimulation also yielded distinct FEP features from both hemispheres (Kruskal-Wallis one-way ANOVA, $F(3,31) = 13.1$–17.8 from the F$_3$-P$_3$, 13.0–15.6 from the F$_4$-P$_4$, $p < 0.005$). In the right hemisphere (ipsilateral to sonication), the use of 0.5 ms PD generated different FEP amplitudes than those obtained from the sham condition in the time segments of 46–49 ms, 93–106 ms, and 130–140 ms (Fig 3B; Tukey-Kramer *post-hoc* analysis $p < 0.005$). The use of 1 and 2 ms PDs generated different FEP amplitudes compared to the sham condition, but in a narrower time-segment compared to those obtained from the 0.5 ms PD (105–118 ms and 55 ms; Tukey-Kramer *post-hoc* analysis $p < 0.005$). From the left hemisphere contralateral to sonication, the use of a 0.5 ms PD generated different FEP compared to the one from the sham condition in 84–100 ms segment (Tukey-Kramer *post-hoc* analysis $p < 0.005$). On the other hand, the FEP amplitudes between the hemispheres as well as between the stimulation targets were equivalent across the PD conditions (S4 Fig). No correlation was found between acoustic intensity (*in situ* I$_{sppa}$) and the averaged peak FEP amplitude (measured during the sonication duration) across the experimental conditions and stimulation targets (correlation analysis, all R$^2 \leq 0.23$, $p \geq 0.51$, S5 Fig). From the assessment of potential sonication-related artifacts in EEG measurement, we did not find differences in EEG time-series amplitudes between the sonication conditions and the sham ($p < 0.005$; S6 Fig.).

## Comparisons between FEP versus AEP

The total number of AEP data (all out of 80) obtained from binaural stimulation, after excluding signal saturation and motion-related artifacts, was 71.3 ± 3.9, 70.6 ± 3.7, 70.1 ± 3.5, and 70.1 ± 3.7 from 1,400 Hz, 700 Hz, 350 Hz, and no-sound conditions, respectively. The equivalent values from the tonal stimulation to the left ear were 70.6 ± 5.2, 70.4 ± 3.9, 70.3 ± 4.2, and 68.4 ± 4.0 while the values were 69.9 ± 5.1, 70.0 ± 5.9, 68.3 ± 6.9, and 68.0 ± 7.0 from tonal stimulation delivered to the right ear. The data selection was unbiased across the experimental conditions (Kruskal-Wallis one-way ANOVA, $F(3,31) = 0.81$, 1.72, and 0.83, $p = 0.84$, 0.63, and 0.84, for both, left, and right ear, respectively).

The group-averaged AEP responding to binaural and monaural tone stimulations (in Fig 4) revealed no distinct peak from any of the conditions. Comparison among the different tonal

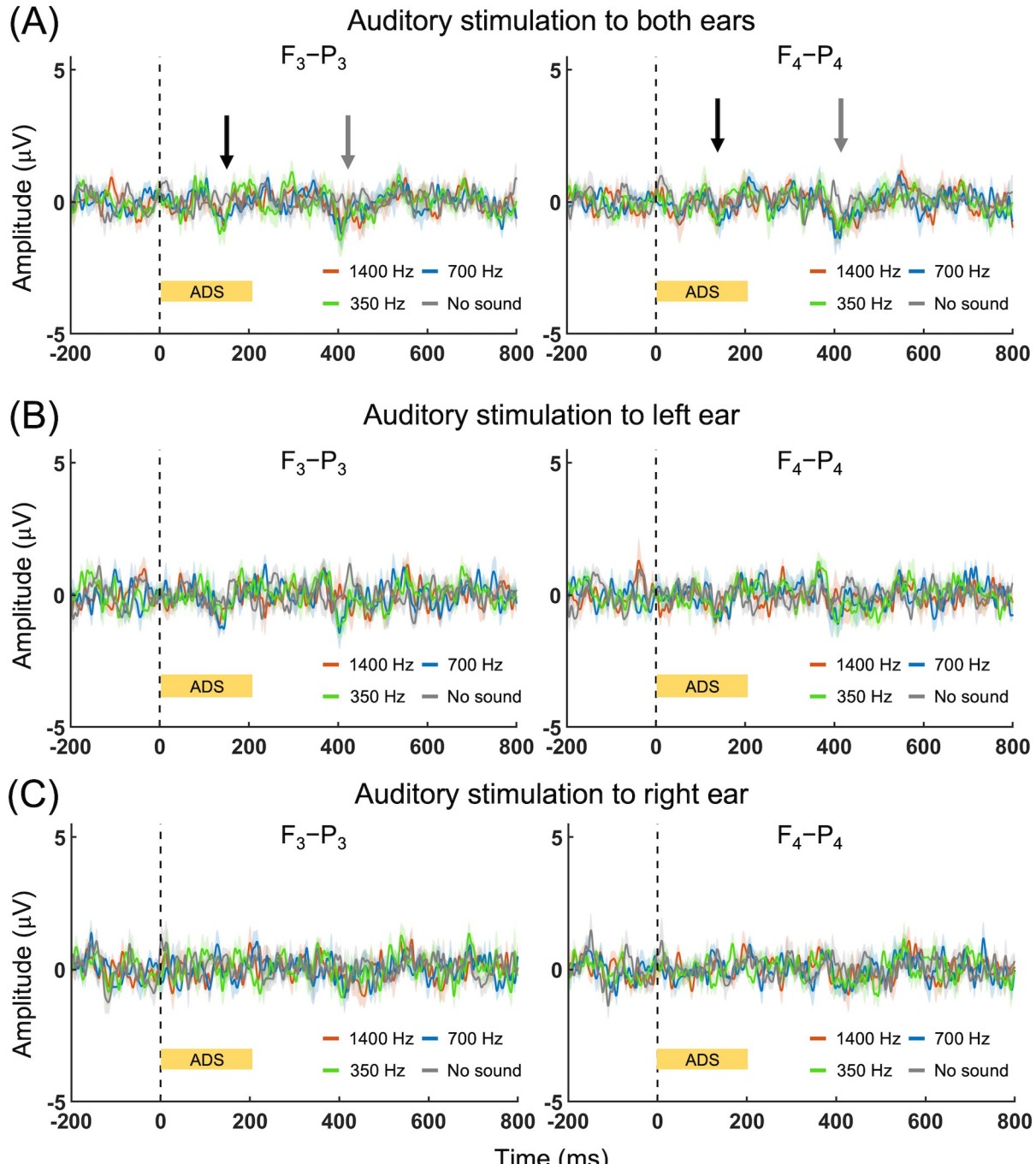

**Fig 4. Frequency-specific comparisons of AEP.** Group-averaged AEP obtained from auditory stimulation of (A) both, (B) left and (C) right ears. Black and gray arrows indicate the timing of weak negative peaks appeared from binaural stimulation. Kruskal-Wallis one-way ANOVA ($p < 0.005$) was conducted to compare the four experimental conditions followed by *post-hoc* analysis for multiple comparisons ($p < 0.005$). No difference was detected. ADS: the duration of auditory stimulation. The solid lines and shaded areas indicate group-average and standard errors, respectively.

conditions (including the no sound condition) did not show any differences in the detected AEPs, except for the presence of weak (~1–1.5 μV) negative peaks having approximately 100 ms and 400 ms latencies observed from binaural stimulation (black and gray arrows, Fig 4A). These weak peaks shared similarity to those obtained from cortical AEP (CAEP) in a clinical setting [51]. Furthermore, AEP features having short latency (< 5 ms), typically detected in clinical diagnosis from A-Z EEG montage (*e.g.*, auditory brainstem responses, ABR) [51], were not observed beyond noise level (*i.e.*, ±1 μV) across all conditions, suggesting lower efficiency of detecting neural response to tonal stimulation using the F-P montage used in the present study.

## rsFC analysis

Comparisons between rsFC maps obtained from Visit #1 (baseline) and Visit #3 (*i.e.*, Visit #3 > Visit #1) showed that FUS stimulation enhanced the rsFC between the sonicated S1 and a cluster of cortical areas in the: (1) pre-/postcentral gyri and paracentral lobule (BA 4/6, contralateral to FUS), (2) supramarginal gyrus posterior to the FUS area and inferior parietal lobule (BA40, ipsilateral to FUS), and (3) middle cingulate cortex (BA24; Fig 5A and Table 2). We also found enhanced rsFC between the sonicated VPL and a wide network of cortical/subcortical areas (Fig 5B and Table 2), which included: (1) insula and putamen ipsilateral to FUS, (2) middle/inferior temporal gyri contralateral to FUS (BA21), (3) precentral and superior frontal gyri, including the supplementary motor area (BA 4/6) ipsilateral to FUS, (4) superior/medial frontal gyrus (BA10) contralateral to FUS, (5) temporal gyrus (BA22 ipsilateral to FUS), (6) inferior frontal gyrus (BA45) contralateral to FUS, and (7)

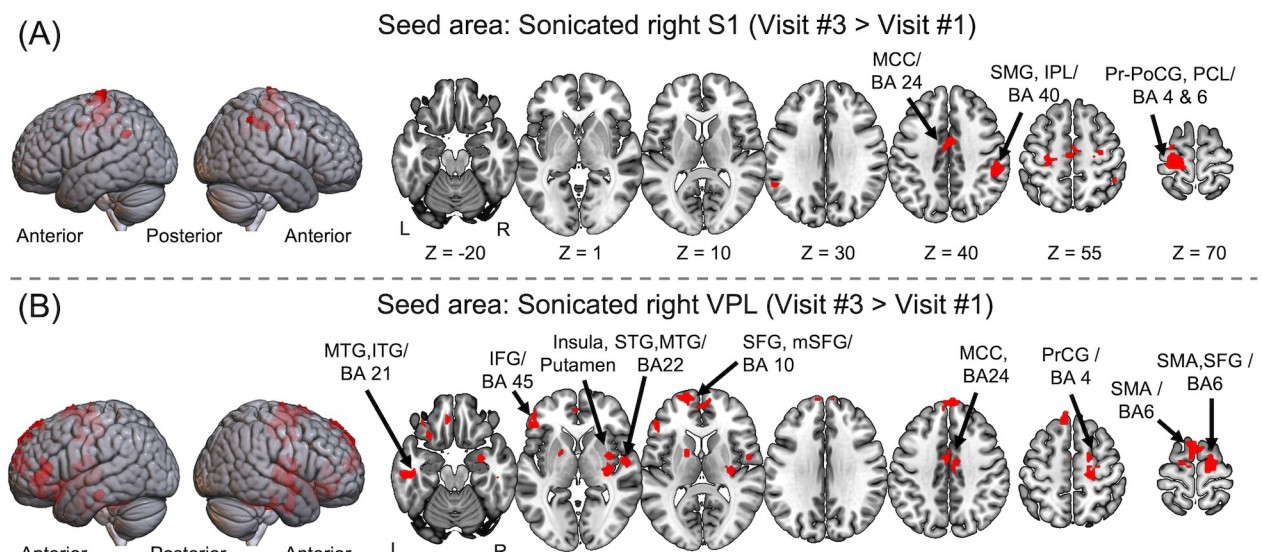

**Fig 5. The brain areas showing greater rsFC with respect to the sonicated (A) S1 and (B) VPL ~1 hour after the sonication (Visit #3 > Visit #1; Bonferroni corrected $p < 0.05$, minimum cluster size > 30 voxels).** A decrease in the rsFC was not detected. L: Left, R: Right, m: Medial, A: Anterior, P: Posterior, BA: Brodmann's area, S1: Primary somatosensory area, VPL: Ventral posterolateral nucleus, MCC: Middle cingulate cortex, SMG: Supramarginal gyrus, IPL: Inferior parietal lobule, PrCG: Precentral gyrus, PoCG: Postcentral gyrus, PCL: Paracentral lobule, MTG: Middle temporal gyrus, ITG: Inferior temporal gyrus, IFG: Inferior frontal gyrus, STG: Superior temporal gyrus, MTG: Middle temporal gyrus, SFG: Superior frontal gyrus, SMA: Supplementary motor area.

**Table 2. Brain regions showing greater rsFC in Visit #3 compared to Visits #1 and #5 (i.e., Visit #3 > Visit #1 and Visit #3 > Visit #5) with respect to the sonicated seed areas, arranged in a descending order in terms of maximum *t*-value within the cluster (Bonferroni corrected *p* < 0.05 with a cluster size of a minimum 30 connected voxels).** The averaged rsFC levels (*i.e.*, fisher's z transformed correlation coefficient, *r*) at the brain regions across subjects were represented as the effect size. No significant change in rsFC patterns was observed comparing Visit # 1 > Visit #3 and Visit #5 > Visit #3.

| Seed area | AAL/BA | Size | Foci in mm (x, y, z) | Peak intensity (t-value) | Effect size (r) | |
|---|---|---|---|---|---|---|
| | | | | | Visit#3 (SE) | Visit#1 (SE) |
| Visit #3 > Visit #1 | | | | | | |
| S1 (R) | PrCG, PoCG, PCL (L), MCC (m) / BA 4, 6, 24 | 326 | -21, -19, 76 | 5.25 | 0.52 (0.15) | 0.12 (0.12) |
| | SMG (L) / BA 40 | 32 | -57, -52, 34 | 4.41 | 0.15 (0.09) | -0.15 (0.06) |
| | SMG, IPL (R) / BA 40 | 102 | 54, -40, 46 | 4.32 | 0.30 (0.10) | -0.01 (0.11) |
| VPL (R) | Insula, Putamen, FG (R) | 230 | 33, -22, 4 | 5.98 | 0.72 (0.19) | 0.44 (0.22) |
| | MTG, ITG (L) / BA 21 | 46 | -54, -22, -20 | 5.73 | 0.26 (0.16) | 0.00 (0.12) |
| | PrCG, SFG, SMA (R) / BA 4, 6 | 210 | 18, -13, 67 | 5.40 | 0.34 (0.11) | 0.09 (0.16) |
| | SFG (L), SFG (m) / BA 10 | 159 | -18, 62, 10 | 5.38 | 0.22 (0.15) | -0.12 (0.15) |
| | STG, MTG (R) / BA 22 | 116 | 51, -7, 1 | 5.03 | 0.46 (0.23) | 0.20 (0.16) |
| | IFG (L) / BA 45 | 120 | -51, 35, -2 | 4.92 | 0.27 (0.15) | 0.03 (0.14) |
| | MCC (m) / BA 24 | 79 | 9, -10, 40 | 4.66 | 0.46 (0.17) | 0.24 (0.11) |
| | SMA (L) / BA 6 | 74 | -3, 8, 70 | 4.14 | 0.29 (0.18) | 0.03 (0.16) |
| Visit #3 > Visit #5 | | | | | | |
| | | | | | Visit#3 | Visit#5 |
| S1 (R) | PrCG, PoCG (L), PCL (L) / BA 4 | 114 | -30, -28, 67 | 4.47 | 0.85 (0.11) | 0.36 (0.07) |
| | Calcarine (L) | 30 | -3, -64, 4 | 4.01 | 0.24 (0.08) | -0.04 (0.08) |
| VPL (R) | PrCG (R) / BA 4 | 62 | 21, -25, 52 | 5.62 | 0.46 (0.19) | 0.13 (0.12) |
| | Insular, Putamen (R) | 83 | 33, -22, 4 | 4.48 | 0.72 (0.19) | 0.52 (0.19) |
| | MCC (m) / BA 24, 32 | 64 | -3, 14, 34 | 4.05 | 0.48 (0.25) | 0.17 (0.17) |
| Visit #1 > Visit #3 & Visit #5 > Visit #3 | | | | | | |
| S1 (R) | None detected | | | | | |
| VPL (R) | | | | | | |

AAL: Automated anatomical labeling, BA: Brodmann area, SE: standard error, L: Left, R: Right, m: Medial, S1: Primary somatosensory area, VPL: Ventral posterolateral nucleus, PrCG: Precentral gyrus, PoCG: Postcentral gyrus, PCL: Paracentral lobule, SMG: Supramarginal gyrus, IPL: Inferior parietal lobule, MCC: Middle cingulate cortex, MTG: Middle temporal gyrus, ITG: Inferior temporal gyrus, SFG: Superior frontal gyrus, SMA: Supplementary motor area, STG: Superior temporal gyrus, IFG: Inferior frontal gyrus, IFG: Inferior frontal gyrus.

middle cingulate cortex (BA24). A decrease in the rsFC was not detected from the contrast, Visit #1 > Visit #3.

From the comparison between rsFC maps obtained from Visit #3 and Visit #5 (*i.e.*, Visit #3 > Visit #5; results summarized in Fig 6A and Table 2), the sonicated S1 region showed a reduced level of rsFC in the clusters of (1) calcarine, and (2) pre-/postcentral gyri and paracentral lobule (BA 4/6) contralateral to FUS stimulation. The location of the pre-/postcentral gyri was approximately the same as the regions that showed greater rsFC from Visit #3 compared to Visit #1. From examination of the rsFC between the sonicated VPL and other brain regions, (1) the precentral gyrus (BA 4) ipsilateral to FUS, (2) insula and putamen ipsilateral to FUS, and (3) middle cingulate cortex (BA24/32) showed lower rsFC in Visit #5 compared to Visit #3 (Fig 6B). No significant increase in rsFC patterns was observed in Visit #5 compared to Visit #3 (based on Visit #3 < Visit #5 contrast). Between Visit #1 and #5 (*i.e.*, comparison between pre-FUS baseline and ~1 month after the thalamic stimulation), no significant change in rsFC was observed (Bonferroni corrected *p* < 0.05, minimum cluster size >30 voxels, Figure not shown).

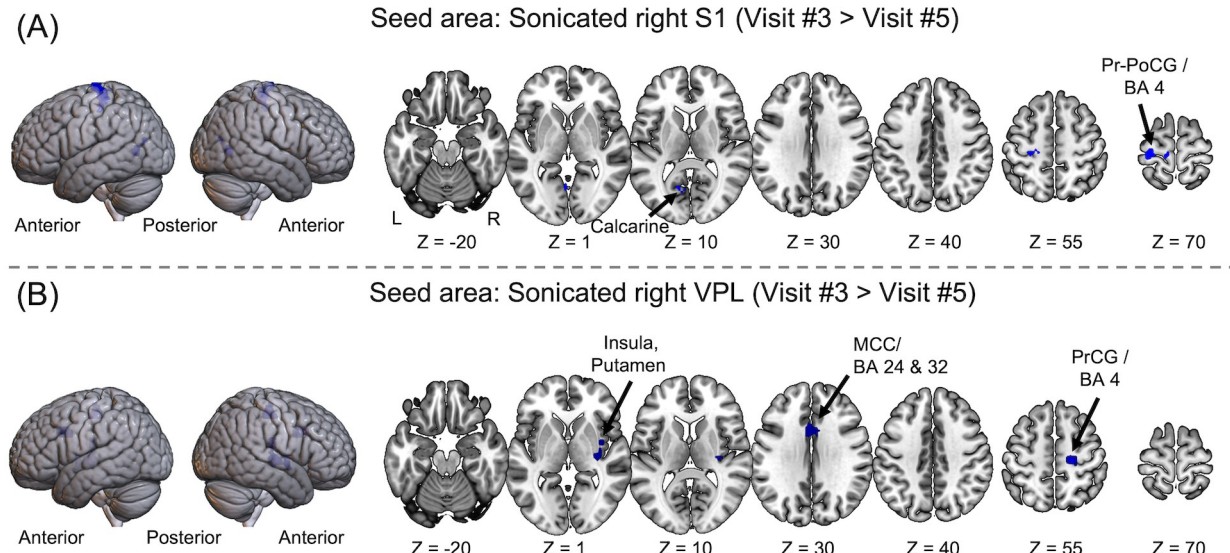

**Fig 6. The brain areas showing reduced rsFC in ~1 month after the sonication (Visit #3 > Visit #5) with respect to the sonicated (A) right S1 and (B) right VPL (Bonferroni corrected _p_ < 0.05, minimum cluster size > 30 voxels).** An increase in the rsFC was not detected. L: Left, R: Right, A: Anterior, P: Posterior, BA: Brodmann area. S1: Primary somatosensory area, VPL: Ventral posterolateral nucleus, PrCG: Precentral gyrus, PoCG: Postcentral gyrus, MCC: Middle cingulate cortex.

### Neurological/neuroradiological/EEG assessment

No cavitation events were detected in any of the FUS sessions, which agree with the premise that cavitation is unlikely to occur at the applied pressure level [52]. No discomfort during and after the sonication was reported from any of the participants throughout the study period. None of the individuals reported any subjective aftereffects of the stimulation or developed any new abnormalities upon NEs (five times), simplified EEG, or neuroanatomical MRI (three times).

## Discussion

### FUS system performance

The semi real-time numerical simulation of sound propagation through the skull allowed for accurate stimulation targeting during the image-guidance phase, with the group-averaged spatial error being smaller than the focal dimension. However, we note the presence of a relatively large spatial error (≥ 3 mm) observed from a few participants (_e.g._, ID #1, #3, and #5 for S1 stimulation and ID #4 for VPL stimulation, shown in Table 1). We surmise that this error was attributed to inadvertent headgear movement which might have occurred after the initial transducer positioning. This calls for improvement in more stable headgear design for reliable targeting. We also encountered difficulties in positioning the FUS transducer to make tight contact with the scalp among a few individuals based on their hairstyle (_e.g._, braided curly hair). Although hairs do not likely to absorb acoustic energy in the low frequency range [53], air bubbles could be trapped between hairs, possibly absorbing acoustic waves or even distorting the acoustic propagation. Shaving hair would not be advised among healthy participants, therefore, on-site validation of the absence of air bubbles in the acoustic path, for example, through ultrasound image-based assessment of the acoustic transmission through scalp [10], can be adopted to ensure air-free acoustic coupling. In addition, we note that the use of a fixed

incident $I_{sppa}$ in the present study yielded individually different *in situ* intensities at the focus, potentially underdosing the thalamic target in one of participants (ID#7, Table 1). To maintain the same *in situ* intensity across all participants, a more rapid, real-time numerical simulation algorithm such as deep-learning-based network or a differential revolution [54, 55] can be used to estimate the derating factor specific to an individual and sonication geometry. Commercial robotic positioning devices, which are available to apply TMS coils in the desired orientation and location [56], may also be conjunctionally used for reproducible FUS stimulation.

## Subjective reporting

The occurrences of perceived tactile sensations from the hand in response to the FUS stimulation were far fewer than the previous observation from visual/somatosensory stimulations (57.9% from visual area stimulation [12] and 91.7% from somatosensory area stimulation [8]). Despite this lower occurrence of tactile sensations, the type of descriptors, such as 'tingling/ vibrating' or 'muscle movement', were congruent with those reported from a previous study on stimulating the somatosensory area in humans [8]. We initially speculated that the marginal perception rate was attributed to inaccurate stimulation location or low *in situ* acoustic intensity; however, no correlations were found between the presence of tactile sensation and the intensity at focus. Thus, we postulate that it might have stemmed from reduced stimulation efficiency associated with the randomized presentation of sonication having different PDs, further confounded by the insertion of no FUS conditions in-between. As the tissue-level sensitivity to FUS can be affected by the types of neurons that have different sensitivities to sonication parameters [57], it is also plausible that the short stimulation intervals used in the present study (4 s) might have decreased neuronal responses to the stimulation. In addition, as sensations perceived during brain stimulation may vary widely in time as well as between individuals (as also observed in case of TMS stimulation [58, 59]), further investigation is needed to evaluate intra- and inter-session reproducibility of FUS brain stimulation and associated perception of sensations, based on the use of a fixed set of sonication parameters.

## FEP comparisons across conditions

Despite the lower rate of tactile perception, FUS stimulation elicited distinct EEG features across all subjects. The acquired FEP shared similarity with SSEP features that were elicited by unilateral median nerve stimulation (*e.g.*, a negative peak appearing with 80–100 ms latencies followed by a positive peak having ~200 ms latencies; Fig 3) whereas the sham condition did not elicit any specific EEG peaks. However, these peaks appeared much later than the those of SSEP (the first negative peak at 51 ms and positive peak at 150 ms upon electrical stimulation). Furthermore, the FEP showed smaller EEG peak magnitudes. We surmise that these differences between FEP and SSEP stemmed from different mechanisms (SSEP reflects afferent brain activity responding to peripheral stimulation whereas FEP detects the neuronal activity caused by external brain stimulation). Despite these differences, these results suggest the involvement of overlapping neuronal pathways between the responses to stimulations. We also note that FEP, unlike electrical/magnetic stimulation of the brain, did not contain any significant stimulation-related artifacts (S6 Fig), which indicates that acoustic stimulation is compatible with studies where minimal EEG artifacts are desired.

The examination of FEP across the PD conditions (Fig 3B) revealed that the use of a 0.5 ms PD generated more profound amplitude differences (compared to the sham condition) than the use of 1 and 2 ms PDs from both S1 and VPL stimulation. The different EP amplitudes were observed from the right hemisphere ipsilateral to sonication ($F_4$-$P_4$ bipolar reading).

These PD-dependent differences in FEP, especially from the use of a 0.5 ms PD, share similarities with the results evaluating the effects of different pulsing schemes in stimulating the brain of large animals (sheep) [60, 61]. The PD-dependent stimulation efficiency has also been demonstrated in our previous rodent study [18] and in studies measuring cell-type responsiveness and activation of mechanosensitive ion channels to ultrasound [57, 62]. This finding suggests the existence of a specific set of sonication parameters that yields higher stimulation efficiency in humans.

## FEP amplitude versus in situ acoustic intensity

We found no correlation between the *in situ* acoustic intensities and the FEP amplitude. It differed from our previous finding which showed a positive correlation between acoustic intensity and the amplitude of EP responses to the FUS stimulation of the sheep visual area [63]. We conjecture that the present observation was attributed to the rather narrow range of tested intensities compared to the ones used in the sheep experiments, which varied up to 10.5 W/$cm^2$. A high rate of inter-subject variability, which has also been observed in large animals, might have contributed to our observation. The use of higher acoustic intensity may increase the amplitude of electrophysiological responses, however, would require caution as it would elevate the risk for mechanical tissue damage or temperature elevation.

## FEP between the hemispheres and sonication targets: Contribution from auditory confounder?

Although the differences between FEP and EEG from the sham condition were distinct from the right hemisphere (ipsilateral to FUS stimulation) compared to the left hemisphere, direct comparison of FEP amplitude between the hemispheres (S4A Fig) and between the sonication targets (the S1 and the VPL; S4B Fig) per PD condition revealed no difference. These results were unexpected since we anticipated more lateralized FEP responses ipsilateral to sonication (*i.e.*, $F_4$-$P_4$), with different features between stimulating the S1 and the VPL. We initially suspected that the observed FEPs were confounded by the auditory sensation experienced by many participants, as similarly being reported as auditory event-related EEG readout associated with transcranial ultrasound stimulation of the primary motor cortex by Braun and colleagues [64]. However, our examination of AEP among separate volunteers (Fig 4) revealed that the presented auditory tones, although delivered in sufficiently high sound intensities aimed to be clearly heard, did not generate detectable EEG potentials. Furthermore, the electrode montage used in the present study (F-P) was not optimized for detecting auditory EEG responses. Therefore, we postulate that auditory perception can be ruled out as a source for the observed bilateral FEP, and instead, the existence of strong interhemispheric functional connections between the somatosensory cortices might have contributed. The SSEP from unilateral median nerve stimulation showed bi-hemispheric EEG peaks (Fig 3A), albeit at a reduced amplitude from the opposite hemisphere, which also partially supports our postulation. However, we note that the measured FEP may also reflect sensory evoked potential to the tactile sensation on the scalp, although only one participant experienced such a sensation (ID#7 under VPL stimulation). Thus, further investigation is needed to isolate the cause for the observed bilateral FEP.

The sensory perceptions during TMS, such as a clicking noise and tactile sensation at the skull interface, are well described [65, 66] and may confound the characterization of neural responses to brain stimulation. Although the degree of sensory stimulation by FUS seems weaker than that of TMS, the present study strongly suggests that FUS is not 'silent' and even may cause weak tactile sensations from the scalp (Table 1). Studies have revealed that FUS can

generate skull-mediated shear waves that may stimulate the cochlea [67] and may elicit auditory sensations during stimulation [47]. The acoustic waves, being enveloped with the same frequency as the PRF (given in audible range), may interact with bone or the inner ear in the path of sonication, creating auditory perception. These interactions will be inevitably proportional to the acoustic intensity. For example, Johnstone *et al.* [47] reported that the application of FUS through the skull inion at a much higher intensity than those used in the present study (16 W/cm$^2$ I$_{sppa}$) produced auditory perception across their entire participant pool (*n* = 7). As the characteristics of sound conduction through the bone may differ from the one used in measuring the AEP, it is not possible to entirely rule out auditory confounds during stimulation. To remedy the auditory confounder, auditory masking [64] (*e.g.*, presentation of audio tones synchronized with sonication or scanner noise from MRI) or amplitude modulation of acoustic waves [68] can be sought after.

### rsFC analysis

Measured about an hour after VPL stimulation and a week after the S1 stimulation (in Visit #3), the rsFC strength increased between the sonicated S1 and a cluster of sensorimotor areas (BA 4 and 6) contralateral to FUS stimulation in addition to the sensory association areas (*i.e.*, IPL; BA40) ipsilateral to the stimulation (Fig 5A and Table 2). The elevated rsFC in these areas, as components in the main sensorimotor circuits, suggests that off-line stimulatory effects of FUS strengthened the level of motor-related connectivity, with extended involvement of the area contralateral to the stimulation. The FEP results showing bi-hemispheric involvement also suggest that FUS stimulation of the unilateral S1 may lead to greater off-line FC changes in the opposite hemisphere. It is also possible that hand dominance (all subjects were right-handed in the present study) might have contributed to the finding as the activation of the somatomotor areas of the non-dominant hand elicits greater bilateral involvement [69]. The elevated rsFC with the anterior portion of the middle cingulate cortex (BA 24) may further indicate the effects of FUS on enhancing the integration of sensory information [70, 71].

   With respect to the sonicated VPL, increased rsFC was observed across a greater extent of brain areas than the S1 (Fig 5B and Table 2), notably in the subcortical putamen and insula, as well as in the sensorimotor areas (BA 4 and 6) ipsilateral to FUS. In addition, contralateral superior and middle temporal gyri (BA22) and the superior frontal gyri (BA10) showed increased connectivity to the sonicated VPL area. These areas are closely related to default circuits for sensorimotor control [72, 73] and are known to be involved in the integration of sensory information (BA22 [74, 75] and BA10 [76, 77]). The elevated rsFC from the middle cingulate cortex and the inferior frontal gyrus (BA45) contralateral to FUS stimulation, matches with the involvement of these brain regions in high-order sensory information processing [78–81] and their mutual connectivity [82]. The observed rsFC enhancement across multiple brain areas also aligns with previous studies showing a network-wide increase in FC responding to TMS/tDCS [83–85].

   The different rsFC level between Visits #1 and #3 indicates that the effects of VPL stimulation outlasted the sonication session beyond ~1 hour (82.5 min). This finding is consistent with a recent study in humans that showed a 14 min administration of repetitive transcranial ultrasound (rTUS) to the primary hand motor area enhanced the TMS-mediated motor evoked potentials for at least 30 min post stimulation [11]. Investigation on primates whereby a short duration (40 s) of ultrasound stimulation of the SMA, although the intensity used was much higher than one used in the present study (> 24 W/cm$^2$ I$_{sppa}$ at 30% duty cycle, > 7.2 W/cm$^2$ I$_{spta}$), also led to the modulation of off-line FC lasting more than an hour after the stimulation [3]. Put together with in vitro evidence of sustained modulation of rodent neuronal

excitability [24], FUS may confer stimulatory effects across the brain for a duration that is sufficient to induce neural plasticity.

When the effects of FUS were compared between Visits #3 and #5 (about 4 weeks after the thalamic stimulation), most of the brain areas associated with the increased connectivity in Visit #3 from Visit #1 showed decreased rsFC in Visit #5 (Fig 6A and Table 2). However, a significant portion of the areas having the increased rsFC in Visit #3 (compared to Visit #1) were not identified in Visit #5 (*e.g.*, BA's 10, 21, 22, and 45). Based on equivalent rsFC between Visits #1 and #5, we anticipated that most of the neural circuits that showed different rsFC between Visits #1 and #3 would have been detected in comparison between Visits #3 and #5. Although the small subject group in the present study would limit the sensitivity of FC analysis, we surmise time-dependent shifts in the FC level might have contributed to this discrepancy. In addition, we cannot completely rule out the possibility that the stimulatory effects of FUS may be incompletely recovered back to the pre-FUS state during Visit #5.

We acknowledge that the present study lacked separate assessment of the degree of normal variabilities in rsFC across the study period, without administering FUS stimulation. Although previous studies have shown that a week time gap does not introduce more variability to rsFC beyond normal inter-session variability across much longer time period (e.g., three weeks [86] or 3.5 year [87]), the time gap between the Visit#1 and #3 (about a week) might have contributed individual fluctuations in the rsFC. Acquisition of baseline rsFC data right before the initial FUS stimulation session may help reduce this potential confounder. We also note that the observed changes in FC in Visit #3 may not be solely related to the VPL stimulation and may include effects from the S1 stimulation (which was given about a week prior). Zhang and colleagues suggested that 7 days may be considered a sufficient washout period after stimulating the primary hand motor area using rTUS [11]; however, it is possible that modulatory effects of the S1 may remain about a week later, being mixed with the effects from thalamic stimulation. Counter-balanced randomization or a crossover between the S1 and VPL stimulation sessions can be sought after; however, considering the small number of subjects in the study, it may introduce potential mutual interactions between the stimulatory effects, which serve as further confounds during the interpretation of the data. In addition, contributions from different PD conditions on rsFC were not examined as subjects underwent the FUS session containing all three PD conditions mixed in a randomized fashion. These limitations call for separate rsFC data acquisitions per stimulation targets, as well as per sonication parameter to achieve more robust rsFC analysis to reveal the therapeutic utility of transcranial FUS.

### Neurological/neuroradiological/EEG assessment

Repeated neurological and MRI examinations revealed that none of the participants had any short- or long-term effects from the sonication. A few studies on large animals/primates have been conducted at much higher intensities than those used in the present study. For example, 20.5 W/cm$^2$ I$_{sppa}$ has been used to stimulate sheep in awake state without generating any brain damage at 70% duty cycle [60] while 51.6 W/cm$^2$ I$_{sppa}$ was safely used to stimulate the primary visual cortex in non-human primates at 50% duty cycle [5]. Although careful evaluation on risk factors such as skull/tissue heating is needed, we believe that FUS given in the current stimulation protocols can be used safely among healthy individuals.

### Technical limitations and future directions

The acoustic focus with elongated shape, especially during VPL stimulation (shown in S3 Fig), may stimulate brain regions other than the designated brain area. A phase-array transducer configuration would be advantageous to create a tighter acoustic focus than a single-element

transducer for deep brain stimulation [88]. Although the present study used a numerical simulation to model acoustic propagation through an individual skull, in vivo mapping of the acoustic focus is also desired to provide its 'ground truth' location and intensity. Moderate heating of the brain tissue and subsequent detection of temperature elevation through the use of MR-thermometry [89] may not be justifiable to be used among healthy volunteers, not to mention that the low acoustic intensity used in the brain stimulation would not raise tissue temperature. The MR-based acoustic radiation force imaging techniques [90, 91], though their sensitivity is not yet sufficient to detect low-intensity FUS, are continuously evolving and may provide a means to image the acoustic focus [92], for example, through MR-elastography that offers ultra-high sensitivity [93]. We also note that the 1-mm slice gap used in rsfMRI data acquisition was rather wide even considering the potential contributions from the signal crosstalk between the slice selection of MRI. Higher spatial resolution, on the order of 2.5 mm$^3$ isovoxel size, and adoption of an imaging sequence such as multiband multi-echo fMRI [94], would improve sensitivity and reproducibility of the rsFC.

Instead of the passive sham condition used in the present study, addition of an active sham control condition, for example, a simple inversion of a FUS transducer facing toward outside of the brain [6, 10], may benefit studies in humans. Advanced active sham control can be sought after by sonicating the brain regions away from the targeted regions [28]; however, the selected target locations to create such a sham condition are limited in the presence of highly extensive functional connectivity in the human brain and would warrant further investigation. Another limitation of the present work is that only the excitatory effects were examined despite the bimodal neuromodulatory capabilities of FUS stimulation [15, 16, 25, 60]. For example, low duty cycle (~5%) FUS that is given for a duration on the order of minutes has shown to impart suppressive effects on somatosensory areas in large animals [19, 60]. Further in-human investigations of pulsing parameters on suppressive effects are warranted, for example, through examination of electrophysiological (*e.g.*, reduction of SSEP magnitude) or behavioral responses (*e.g.*, two-point discrimination tasks [95]).

## Conclusions

Transcranial FUS can safely stimulate both cortical and thalamic somatosensory circuits in humans and elicits EEG responses that are accompanied with enhanced rsFC across a network of brain areas. The offline effects of thalamic stimulation persisted at least one hour, which suggests promising potential for inducing neural plasticity that is necessary for various therapeutic interventions. The generalization of our findings on rsFC was limited by the small sample size (*n* = 8), rendering conclusive derivation of the effects size and power analysis difficult to ascertain. Although the central tendencies of functional brain organization among different experimental groups can be captured from a group of ~24 subjects [96, 97], lack of information regarding normal inter-subject/inter-session variabilities of rsFC data in the context of FUS-mediated brain warrants further investigations involving a greater number of study participants. As FUS stimulation may accompany auditory perception that can impact the interpretation of neural responses, further remedial countermeasures are also urgently needed.

## Supporting information

**S1 Fig. An experimental setup for the assessment of the fidelity of acoustic simulation.** (DOCX)

**S2 Fig. The spectra of passive cavitation signals measured up to the 5th harmonics (1250 kHz) of FUS sonication.**
(DOCX)

**S3 Fig. Spatial profile of acoustic intensity obtained from numerical simulation.**
(DOCX)

**S4 Fig. Group comparison FEPs between hemispheres and stimulation targets.**
(DOCX)

**S5 Fig. The comparison between *in situ* $I_{sppa}$ and FEP amplitude across different pulsed durations, including the sham condition.**
(DOCX)

**S6 Fig. Examination of sonication-related artifacts in FEP acquisition.**
(DOCX)

**S1 Table. The difference between the simulation and actual measurement of the FUS focus.**
(DOCX)

**S1 Appendix. Validation of numerical simulation.**
(DOCX)

## Acknowledgments

Authors would like to thank the editorial help from Dr. Kavin Kowsari and Mr. Jared Van Reet.

## Author Contributions

**Conceptualization:** Seung-Schik Yoo.

**Data curation:** Hyun-Chul Kim, Wonhye Lee, Seung-Schik Yoo.

**Formal analysis:** Hyun-Chul Kim, Seung-Schik Yoo.

**Funding acquisition:** Seung-Schik Yoo.

**Investigation:** Hyun-Chul Kim, Wonhye Lee, Daniel S. Weisholtz, Seung-Schik Yoo.

**Methodology:** Hyun-Chul Kim, Seung-Schik Yoo.

**Project administration:** Seung-Schik Yoo.

**Resources:** Seung-Schik Yoo.

**Software:** Hyun-Chul Kim, Wonhye Lee, Seung-Schik Yoo.

**Supervision:** Seung-Schik Yoo.

**Validation:** Hyun-Chul Kim, Wonhye Lee, Daniel S. Weisholtz, Seung-Schik Yoo.

**Visualization:** Hyun-Chul Kim, Wonhye Lee, Seung-Schik Yoo.

**Writing – original draft:** Hyun-Chul Kim, Wonhye Lee, Daniel S. Weisholtz, Seung-Schik Yoo.

**Writing – review & editing:** Hyun-Chul Kim, Wonhye Lee, Daniel S. Weisholtz, Seung-Schik Yoo.

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
