## [Decision Letter · Decision Letter 0]

5 Feb 2023

PONE-D-23-00728Transcranial focused ultrasound stimulation of cortical and thalamic somatosensory areas in humanPLOS ONE

Dear Dr. Yoo,

Thank you for submitting your manuscript to PLOS ONE. After careful consideration, we feel that it has merit but does not fully meet PLOS ONE’s publication criteria as it currently stands. Therefore, we invite you to submit a revised version of the manuscript that addresses the points raised during the review process.

Both reviewers have significant concerns that need to be addressed. Please note that data should be publicly available unless there specific reasons which should be specified.

We look forward to receiving your revised manuscript.

Kind regards,

Robert Chen

Academic Editor

PLOS ONE

5. Please include your tables as part of your main manuscript and remove the individual files. Please note that supplementary tables (should remain/ be uploaded) as separate "supporting information" files.

Reviewers' comments:

Reviewer's Responses to Questions

**Comments to the Author**

1. Is the manuscript technically sound, and do the data support the conclusions?

Reviewer #1: No

Reviewer #2: Partly

2. Has the statistical analysis been performed appropriately and rigorously? 

Reviewer #1: Yes

Reviewer #2: Yes

3. Have the authors made all data underlying the findings in their manuscript fully available?

Reviewer #1: No

Reviewer #2: No

4. Is the manuscript presented in an intelligible fashion and written in standard English?

Reviewer #1: Yes

Reviewer #2: Yes

5. Review Comments to the Author

Reviewer #1: This study examines the online and offline effects of focused ultrasound stimulation of the somatosensory cortex and VPL nucleus of the thalamus, in humans. The use of a specially designed headgear for transducer positioning, individual simulations based on CT scans, real-time image guidance and testing of auditory evoked potentials (AEPs) - albeit in a separate participant group - are some strong points of this paper. However, at this point, the reported physiological effects are not convincing, and I have some major concerns about the study.

My main concern is regarding the timeline of measurement of the offline effects.

1. The baseline resting state functional connectivity (rsFC) measurement is made several days before the post FUS measurements. Can you provide any evidence that rsFC measurements are stable and reliable over such long periods? This is particularly important given the small sample size (n = 8).

2. What is the rationale for making the post FUS rsFC measurement 1 hr after VPL sonication, but 1 week after S1 sonication. In fact, on line 670, the authors mention that any FUS effects may be washed out after a week. This makes any reported FUS effects on S1 questionable. Also, why was the sonication of S1 and VPL not counter-balanced?

Further comments –

1. Line 25 – FUS induced or evoked? This is repeated several times in the paper. In EEG literature, “induced” typically refers to a non-phase-locked change in e.g., oscillatory brain activity, whereas “evoked” refers to phase-locked responses (such as classical ERPs)

2. Line 73 – offline effects are still transient. They are not permanent.

3. The fMRI images have relatively low resolution with gaps – how does this affect the connectivity estimates?

4. Please explain the cavitation detection in further detail – does the described method detect actual occurrence of stable vs inertial cavitation or assesses the probability of cavitation? Does it provide any information about potential location of cavitation? Is there invitro evidence that this particular setup is able to pick up stable/inertial cavitation?

5. Line 553 – what do you mean by ‘perception toward brain stimulation’?

6. Line 566 – how did you confirm that the EEG did not contain stimulation-related artifacts?

7. While it is helpful to show that sound alone did not evoke AEPs in the used electrodes, the US is largely bone conducted and may have a different quality. Therefore, it is not possible to entirely rule out auditory confounds, even if the sounds used for eliciting AEPs are loud enough. This should be mentioned as a limitation.

8. For participants who experienced a tactile/ vibratory sensation on the scalp – the FEP may in fact be a sensory evoked potential. Please mention this as a limitation.

Reviewer #2: In this study, Kim et al. investigate the effects of transcranial focused ultrasound stimulation (FUS) on the thalamus (ventral posterolateral nucleus) and the somatosensory cortex. They evaluate ultrasound-evoked potentials and offline effects of FUS on resting state functional connectivity. Auditory evoked potentials by a tone are compared to FUS evoked potentials to rule out the possibility that FUS-mediated effects might be confounded by stimulating the auditory system. This is an interesting report, well written and easy to follow. I have only a few questions and suggestions that the authors may want to consider:

- The paper is limited by its small sample size, which calls into question the generalizability of the results. I suggest the author clearly mention this limitation in the discussion. How large are the effect sizes in MRI resting data results? I am not asking the authors to scan more participants for the current study, but estimation of power and effects size would help other researchers who plan to run similar experiments in future.

- The study could have benefited greatly from having an active sham control. Please add this limitation to the discussion.

- Please provide more information on how spatial errors due to mislocalization of targets were calculated. How robust are these measurements in individual participants? How computationally efficient and time consuming are the numerical calculations? Please add this information to the method section.

- I am wondering whether the targets, especially the thalamus, might be underdosed? What was the rationale for choosing the intensities (14.7 and 9.1 for S1 and thalamus respectively, in free water). Please address this issue in the discussion.

- It is interesting that the participants subjectively reported auditory or somatosensory experiences. I am wondering whether the participants were instructed to expect such sensations or not? Did any participant report those sensations during the sham condition?

- Maybe I have missed it, but although the SSEPs and FEPs look very similar, at least by eyes, it would be more appropriate to directly test whether they are statistically different or not. Furthermore, I am not entirely sure whether one can conclude that since SSEPs and FEPs are similar, FUS has activated a common pathway … (Line 565). This is likely but not a fact though. I suggest to tone down this statement.

6. PLOS authors have the option to publish the peer review history of their article (what does this mean?). If published, this will include your full peer review and any attached files.

Reviewer #1: No

Reviewer #2: No

---

## [Author Response · Author response to Decision Letter 0]

29 Mar 2023

We thank for granting us the opportunity to revise our submission. The Reviewers’ comments were very helpful toward improving the manuscript. We have addressed each point raised by the Reviewers and have listed the responses in the "Response to reviewers" file.

---

## [Decision Letter · Decision Letter 1]

3 Jul 2023

Transcranial focused ultrasound stimulation of cortical and thalamic somatosensory areas in human

PONE-D-23-00728R1

Dear Dr. Yoo,

We’re pleased to inform you that your manuscript has been judged scientifically suitable for publication and will be formally accepted for publication once it meets all outstanding technical requirements.

Kind regards,

Robert Chen

Academic Editor

PLOS ONE

Additional Editor Comments (optional):

Reviewers' comments:

Reviewer's Responses to Questions

**Comments to the Author**

1. If the authors have adequately addressed your comments raised in a previous round of review and you feel that this manuscript is now acceptable for publication, you may indicate that here to bypass the “Comments to the Author” section, enter your conflict of interest statement in the “Confidential to Editor” section, and submit your "Accept" recommendation.

Reviewer #1: All comments have been addressed

Reviewer #2: All comments have been addressed

2. Is the manuscript technically sound, and do the data support the conclusions?

Reviewer #1: Yes

Reviewer #2: Yes

3. Has the statistical analysis been performed appropriately and rigorously? 

Reviewer #1: Yes

Reviewer #2: Yes

4. Have the authors made all data underlying the findings in their manuscript fully available?

Reviewer #1: Yes

Reviewer #2: No

5. Is the manuscript presented in an intelligible fashion and written in standard English?

Reviewer #1: Yes

Reviewer #2: Yes

6. Review Comments to the Author

Reviewer #1: I thank the authors for their revision. All of my previous comments have been addressed sufficiently.

Reviewer #2: (No Response)

7. PLOS authors have the option to publish the peer review history of their article (what does this mean?). If published, this will include your full peer review and any attached files.

Reviewer #1: No

Reviewer #2: No

---

## [Editor Report · Acceptance letter]

12 Jul 2023

PONE-D-23-00728R1 

Transcranial focused ultrasound stimulation of cortical and thalamic somatosensory areas in human 

Dear Dr. Yoo:

I'm pleased to inform you that your manuscript has been deemed suitable for publication in PLOS ONE. Congratulations! Your manuscript is now with our production department. 

Kind regards, 

on behalf of

Dr. Robert Chen 

Academic Editor

PLOS ONE